# The Role of IoT Devices in Sustainable Car Expenses in the Context of the Intelligent Mobility: A Comparative Approach

Javier Goikoetxea-Gonzalez, Diego Casado-Mansilla and Diego López-de-Ipiña *

Facultad de Ingeniería, Universidad de Deusto, 48007 Bilbo, Spain; javier.goikoetxea@opendeusto.es (J.G.-G.);
dcasado@deusto.es (D.C.-M.)
* Correspondence: dipina@deusto.es

**Abstract:** Connected cars have often been defined as vehicles that can provide some services and information without human intervention. Several scholars have examined the factors that promote the purchase or adoption of such augmented vehicles. However, little emphasis has been placed on the determinants for reducing car expenditures when a driver owns a car with an Internet of Things (IoT) device or a smart assistant in the context of smart mobility. Therefore, this article analyzes whether emerging technology such as IoT plays a key factor for a driver concerning the expenses related to the car (e.g., insurance, maintenance, and repairs). To this extent, a methodology based on exploratory (i) and confirmatory analysis (ii) was carried out. The authors initially conducted an exploratory phase by means of a Delphi method in which a group of vehicle experts ($n = 25$) were recruited to give their opinions and reach an agreement defining the determinants that they believed affected vehicle expenditures the most. Secondly, and taking into consideration that the salient determinant from the Delphi method was the use of technology and the warnings and alerts it triggers, a questionnaire was delivered to 556 drivers to analyze the everyday spending on their cars. Specifically, the survey aimed to compare the responses of people who own connected cars or have any kind of built-in IoT infrastructure ($n = 302$) with those of people with non-connected cars ($n = 254$). The main conclusion obtained for this latter approach was that drivers with a connected car have remarkably lower car expenses than those driving a conventional car.

**Keywords:** connected car; IoT; determinants; Delphi method; survey; car expenditures

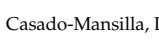



## 1. Introduction

Over the last few years, the automotive domain has been evolving rapidly. One of the different challenges vehicle manufacturers are tackling is the transformation of their innovation approach from offering efficient engines to increasingly offering more software-driven machines. The beginning of a new revolution in the automotive industry has just started, when the concept of technology-connected cars will transform the vehicle completely [1]. The challenge does not lie any longer in waiting for manufacturers to develop new car models equipped with sufficient technology, but rather in embracing the opportunity to transform a conventional vehicle into a connected car through the installation of technological devices [2]. The increase in technological development and the globalization of markets, and the creation of the concept of smart cities and economies are significantly changing the environment in which companies operate. This changing landscape in companies is proof of the current progressive adherence to information technologies (IT), telecom, and even robotics.

Internet of Things (IoT) devices embedded in vehicles can play a relevant role as they can help to transform our cities. Thus, smart mobility is one of the pillars for the design of the new smart cities. If vehicles become smarter through IoT devices embedded into them, it could be possible to link vehicles with infrastructure. This infrastructure

could help us to create more sustainable cities, saving costs and $CO_2$ emissions for citizens and governments.

A connected car is defined by Coppola and Morisio [3] as "a vehicle capable of accessing the Internet, of communicating with smart devices as well as other cars and road infrastructures, and of collecting real-time data from multiple sources". The vehicle, therefore, ceases to be a system that only transports us and becomes a means of transport that can help us satisfy our daily needs. A vehicle may cover our needs in real time by providing the most accurate products and services available at that time. This technology must be able to control some parameters of the car and interact with the vehicle and its driver. This underlying technology is recognized as IoT.

There are diverse technologies that can be used to instrument a car, such as a vehicle communications system (Vehicle to Vehicle—V2V, Vehicle to Infrastructure—V2X, Emergency Call—E-Call), cellular GSM technologies (SIM car with 3G, 4G, 5G), different software associated with car systems (Android and IOS systems, Amazon Car, Voice Assistance) and Internet of Things devices (IoT). An IoT device, based on GSM and Global Positioning System (GPS) technologies, which also takes measurements of the car, e.g., fuel consumption and other parameters through OBDII protocol, was used in this paper to transform a conventional car into a connected car. We used an OBD-II device (On Board Diagnostic)—a standard device connected to the vehicle, that is provided with an internal SIM card that transmits mobility information (e.g., location, speed, acceleration, braking, etc.), as well as information from the car's on-board computer (e.g., engine failure, engine temperature, maintenance needs, etc.).

As an additional description and according to Swan [4], "the connected car means a continuously Internet-connected car, generating and transmitting data". These data can prevent fatigue, provide real-time assistance in case of accidents, and assist cars in performing remote diagnostics.

According to a study by PriceWaterhouseCoopers [5], 40% of total mileage will be made by autonomous vehicles in the European Union (EU) by 2030. In that same year, the pool of vehicles is also expected to fall from 280 million to 200 million. This represents a 28.5% reduction in the passenger-car pool. Thus, we are witnessing a change of paradigm because there will be less demand for vehicles, but they will have to last longer, so the maintenance of the vehicle becomes paramount. This trend is also confirmed by the maintenance cost of electric vehicles, which is much lower than that of petrol cars. An opportunity is, therefore, arising for the aftermarket car industry (car maintenance, repairs, etc.) as vehicles will be responsible for more of the activity transforming the structure of our cities as well. Furthermore, according to the same report, vehicle manufacturers are gradually becoming software development companies to ensure that the needs of the autonomous car are met. In addition to this, the world of the connected car is carving a niche in the evolution of a more technology-based society in terms of mobility and smart cities. In essence, this change of paradigm toward longer-lasting cars will have an impact on the way people maintain their cars. In economic terms, as an average, and following the internal figures of Grupo Next Mobility Company, (www.gruponext.es) it is estimated that in the EU, a passenger car costs around EUR2000 per annum. Any variation below this would suggest significant savings for the drivers and for the environment, since better or reduced driving behavior leads to reduced pollution. Indeed, the revision of some basic elements of the vehicle (oil, filters, tires, etc.) helps to reduce the amount of consumption of the car itself, and, therefore, a well-maintained vehicle helps to reduce $CO_2$ emissions [6].

In conclusion, embedding technology in the vehicle should be the catalyst for developing on-demand service models. By equipping cars with connected car technology, it is becoming easier today to control and determine how much a vehicle drives, how it is driven, what inspections it needs, or how much it spends on petrol when detecting the vehicle parked next to the gas station pump hose. This is a reality now and not only what is going to happen in the future. The emergence of the connected car, therefore, will help to increase the safety of the car and its occupants and will provide more efficient (in terms of

cost and pollution generation) driving, being the prelude to future autonomous driving [7] and the transformation from the "old mobility" to the "new mobility". In terms of giving some context to the reader, the difference between a connected car and an autonomous car is that the first one reads the internal sensors and has connections with the infrastructure and the second drives automatically by sensing the surrounding environment and making driving decisions.

As technology (car instrumentation) can play an important role for drivers, this article examines the role of introducing this technology around car expense management. For this objective, a participatory survey was sent to a group of experts (*n* = 25 at the beginning, but only 22 remained after the full study—thus, three people left the study or did not complete it in its entirety) and followed up with a Delphi method [8,9]:

1.  The objective of the Delphi method was to identify which determinants they believed had the highest impact on the money that drivers spent on their cars. Once these determinants were ranked by all the expert participants and an agreement reached following various rounds of the Delphi method, it was determined that technology has a salient role in forecasting drivers' expenditure on their cars, outperforming other determinants such as marketing campaigns or previous knowledge of the driver.
2.  To confirm the outcomes from the exploratory analysis, a questionnaire was created to obtain evidence from everyday drivers. The survey was conducted to evaluate and compare the spending level of drivers who own a connected car with regards to those who have non-connected conventional cars. A sample of 556 drivers was gathered, where the drivers were divided into two groups: people with assistive technology in their cars (connected cars) and drivers of conventional cars without such technology incorporated.

After the comparison of the responses of the two sample groups, it was possible to answer the main research question of this article: "Does technology play a key role when it comes to reducing the everyday expenses around the car?" To be more precise and for the sake of avoiding misunderstandings, when the authors refer to car expenses, they are talking about the following costs: petrol, insurance, maintenance, car service, garage, tolls, penalties, taxes, and repair. The cost associated with the connected car system, including its hardware (IoT device) and associated software (platform), was not considered, since their costs are negligible compared to all the other car running costs. Besides, drivers adopting them are not charged for them.

Hence, this paper aims to validate the following associated hypothesis: "Connected cars through emerging technology reduce car expenditures in comparison with non-connected cars."

The article is organized as follows: First, we review the relevant literature in three different categories related to the domain of this paper: (i) connected car services, (ii) variables affecting car usage, and (iii) the role of technology in reducing costs. Second, the Delphi method and an in-depth follow-up survey used to conduct an exhaustive study on driver behavior regarding car expenditures are described. Third, the paper includes the results of both studies and gains insights from their analyses. Finally, the paper draws some conclusions and suggests further areas for study.

## 2. Literature Review

This section reviews the variables that could affect car consumption and the technologies that could play an active role in reducing expenses around the car. Figure 1 depicts schematically how the literature review process was carried out. As can be observed, three categories of analyses were performed:

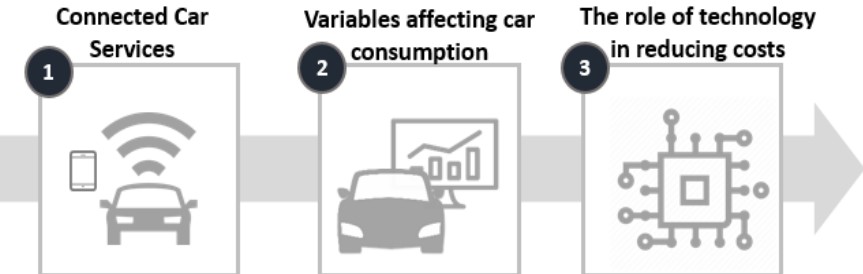

**Figure 1.** Literature review framework.

### 2.1. Connected Car Services

The field of the connected car is part of the world's evolution. In this sense, the introduction of the personal computer (PC) in the field of domestic use could be considered the first step and, subsequently, the internet revolution was the second. Today's world is hyper-connected, and the car is no exception [4]. As Swan [4] also indicates, the greatest limitation of a connected car is data privacy management. In the same context, all the humongous amounts of data could be used to generate different business models [10].

Data are fundamental to value creation and present a competitive advantage in the smart domain [11,12]. All data collected and processed from the car can be quite valuable if they are shared properly [13]. With the data, a lot of services could be generated around the car [14].

There is another important implication of connected cars in the management of the city. They could be seen as part of a system that can be shared with the local authorities in order to send alerts during certain scenarios, such as possible crashes or significant car failures [15]. Datta et al. [15] indicate that the connected car system can be used to develop a V2X (Vehicle-to-Everything) connection, to monitor car mobility and environmental conditions, and to deploy different services around the mobility ecosystem.

On average, 20% of new car buyers state that they would be willing to switch to a different car brand for better connected car services [16]. Mikusz and Herter [16] define car services as services based on the intelligent connection of the vehicle with its environment, such as transportation infrastructure, other vehicles, and drivers. However, we should not forget that all these services need to be relevant to the end user. Hebeler and Hofmann [17] argue that many connected car services are put in place without monetary value for the customers.

The term "connected car" is often associated with applications for better managing car maintenance and services, such as tires changes. [15]. The evolution of the connected car, taking into account technology maturity levels, driving factors, and the business models of connected cars, has been deeply studied by Möller and Has [18].

### 2.2. Variables Affecting Cars' Expenses

Few earlier scholars have been interested in exploring this domain. There are many variables that can play a significant role affecting cars' expenses [19]. Fuel consumption is one of the most relevant costs in car use. Parry [20] indicates that a 9.1% reduction in fuel consumption can be achieved because of embedded technology in the vehicle. Thus, information and communications technology (ICT) may work as an element of persuasion, transforming drivers into more informed people who are therefore sensitive to spending more wisely.

Regarding what can be considered car expenses and how they vary, Kim, Hwangbo, and Kim [21] indicate that expenditures may vary depending on the type of driving, the driving circumstances, the type of car, the speed at which it is driven, and other factors related to car use. There are other studies [22] that show that 44.33% of buyers of electric or hybrid vehicles are also buyers of home solar panels. A total of 12.67% indicate that they have not yet bought solar panels, but they are planning to buy them soon. If both percentages are aggregated, it can be deduced that 57% of the people who drive a

sustainable vehicle (in environmental terms) are also eco-driven in other activities of their ordinary lives. This is a relevant factor since technology can help to condition the attitudes of people, and people who adopt an attitude as a pattern of behavior evolve according to that pattern [23].

*2.3. The Role of Technology in Reducing Price and Environmental Costs*

The existing literature mainly focuses on measuring some car parameters for the purpose of car activity control [24]. Some authors have focused on car maintenance control through radio frequency systems installed on roads. The ability to check a car remotely through technology (e.g., change of oil, change of tires, or change of brake pads) is part of what is being sought in this literature review. According to car manufacturers, managing a vehicle on time always has an impact on direct and indirect cost reduction since a possible deterioration is expected if the vehicle is not subject to the necessary maintenance tasks on time. In this way, Lin et al. [25] mention in their research that with the introduction of remote on-line diagnostic systems connected to a car, the time of fleet management and repairs can decrease significantly.

There is a current trend that is looking at the impact of selling mobility instead of cars. According to an empirical analysis by Firnkorn and Müller [26], private vehicles were reduced as a result of a consumer reaction. This theory confirms that a vehicle is observed as an expense generator and there are, according to the same study, some users who want to pay only for their mobility services. Other studies have reflected on the link between the use of technology and energy [27], and between the management of energy efficiency using technology and gamification [28]. Until now, no written evidence has been found concerning the scope of car use and savings management.

Apart from this, different authors point out that it is necessary to further the analysis of the types of maintenance service that can be offered to a driver in real time owing to the adoption of new technologies (e.g., [29]). This call for research is focused on the management of automatic calls for car maintenance and repairs [25], the collection of vehicle data for new services [30], and new business models for the automotive world [31]. In summary, having reviewed the state of the art in this field, this paper aims to fill a gap in existing mobility literature regarding the identification of what variables are most important concerning car expenditures, and the paper shares the results in an empirical analysis of the variables.

**3. Methodology**

Two different, yet interwoven, methodologies were applied to help answer the research question of this article: on the one hand, (i) a Delphi method was used to identify which determinants a group of experts believe had the most impact on the money car drivers spend on their cars; and on the other hand, (ii) a survey was conducted to analyze whether two population samples, one with connected cars and another without any technology installed in their cars, develop different conducts in terms of how much money they spend on their cars, or not, and the confidence they have in the answers given. In essence, the results of the exploratory Delphi methodology provide information on experts' opinions and trends in car expenditures [32]; the results of the survey with everyday drivers provide confirmatory information about what is happening in a real scenario and validate the findings provided by the experts. Next, the two approaches to identify determinants and then confirm the validity of the most important one are explained.

*3.1. First Phase: Delphi Method Adapted within a COVID-19 Context*

The study was structured into six phases, as can be seen in Figure 2. This methodology was planned and implemented during the COVID-19 pandemic when face-to-face individual or group interviews were not able to be carried out. Hence, a decision was made to adapt this methodology to the new reality at the time of study. Therefore, we were inspired by Sawhney et al. [33], who already used an adaptation of the Delphi method

in their research because of the COVID-19 pandemic, to conduct our research method. The Delphi method is a process used to arrive at a common group opinion or decision by surveying a panel of experts. Those experts respond to several rounds of questions, and the responses are aggregated and shared with the group after each round. The experts can adjust their answers each round, based on how they interpret other group members' responses. The process concludes with a normal consensus about what the group of experts thinks [34]. Since the Delphi method was launched in the COVID-19 period, the authors have used virtual meetings through Zoom, emails, and phone calls to facilitate the dialogue with experts.

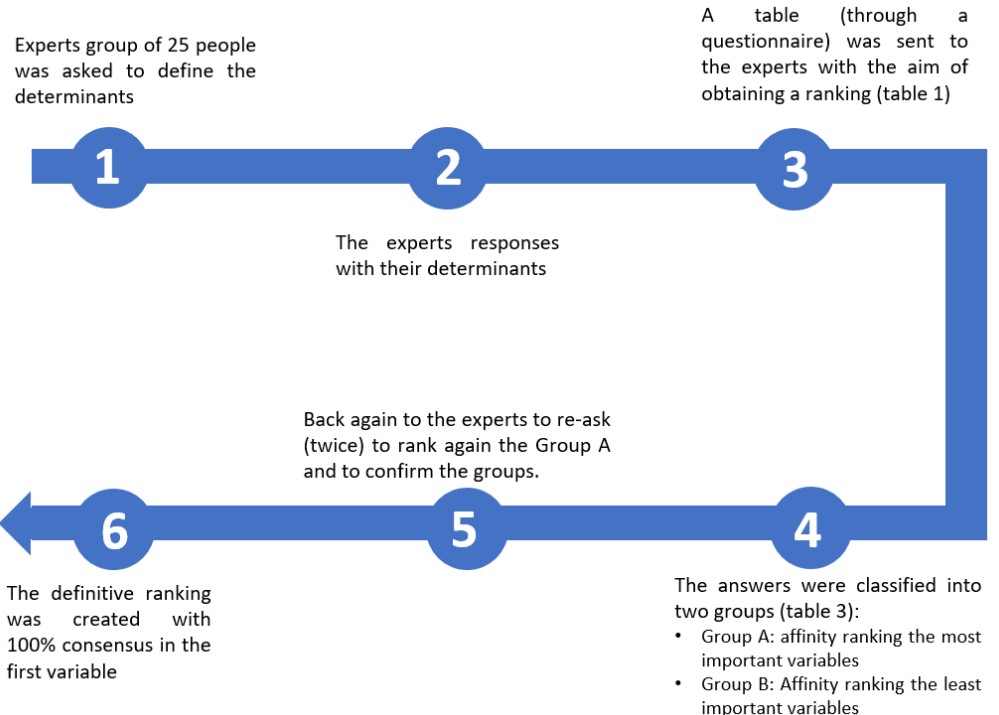

**Figure 2.** Delphi method description along with its six phases.

The following six phases of Figure 2, explained below, were pursued:

A group of 25 experts, in different fields, were selected to explore what determinants would have the greatest effects on their car expenditures from different perspectives (e.g., engineers, behavioral scientists, market analyzers, etc.). The experts were selected from different fields of the automobile world, digital transformation, insurance, garages, and digital marketing. The criteria for selecting the experts considered the idea of covering as wide a range of knowledge as possible. This was possible through collaboration with Grupo Next, a mobility company. Its CEO, a co-author of this paper, assisted in the selection of the experts based on his long experience in the mobility industry (more than 25 years). Thus, of the 25 experts, four are experts in digital marketing, three in the insurance industry, three in the world of connected mobility, two in customer services, three are university members who are experts in digital transformation, three are experts in data processing, two are experts from the OEM aftermarket world, three are experts from the automotive industry and two are experts from the financial world and digital payment services. All are high-ranking executives, i.e., high-ranking executives in their organizations. A determinant, in the context of this research, is defined as a factor or construct with a set of characteristics that defines the spending behavior of a driver around the car. In this sense, a personalised email was sent to each expert. Each expert was asked to indicate the determinants that they believe are the most relevant in terms of influencing a driver's behavior reactions involving car expenses. They were asked to give a minimum of five determinants. The format of providing the information was open and unstructured. Thus, they were asked to reply to

the email by only listing their top selection of determinants associated with a description of them.

Through an on-line Google Form, experts were provided with Table 1 (see the results section) and asked to rank the list of determinants according to their criteria (see Table 2). To avoid any bias in the responses, the determinants were presented in a random order to each expert. It was explained that the higher a determinant was placed in the list, the more relevant that determinant was for the expert in terms of implying the highest impact on drivers' expenses, according to the expert's opinion. On the other hand, assinging a ranking of 10 meant that such a determinant had the lowest impact on car expenses.

**Table 1.** Experts' responses grouped together by the 10 most prominent determinants.

| Determinant No. | Determinant Description |
|---|---|
| Determinant 1 | On-board technology with warning messages generated by the system could influence decision-making over car expenses (showing potential failures and maintenance reminders). |
| Determinant 2 | Promotions launched by Original Equipment Manufacturers (OEM) and the automotive industry could influence decision-making for vehicle maintenance and expenditures. |
| Determinant 3 | Marketing, communications, and promotions are elements that could actively influence a driver's maintenance expenditure decision-making. |
| Determinant 4 | Marketing campaigns generated by automotive brands and dealerships could influence maintenance expenditure decision-making. |
| Determinant 5 | Point-of-sale distance could actively influence drivers' maintenance expenditure decision-making. |
| Determinant 6 | The vehicle's type of use (professional or personal, weekday or weekend) could actively influence drivers' maintenance expenditure decision-making. |
| Determinant 7 | Apps and websites in the field of cars could actively influence drivers' maintenance expenditure decision-making. |
| Determinant 8 | Recommendations from family or close friends could actively influence drivers' maintenance expenditure decision-making. |
| Determinant 9 | Comments and recommendations from friends in the automotive field could actively influence drivers' maintenance expenditure decision-making. |
| Determinant 10 | New transport policies (e.g., 30 km/h zones) and their impact on the environment could actively influence on drivers' maintenance expenditure decision-making. |

**Table 2.** Experts' determinant ranking.

| Averaged Order | Determinant No. |
|---|---|
| 1 | Determinant 4 |
| 2 | Determinant 5 |
| 3 | Determinant 2 |
| 4 | Determinant 3 |
| 5 | Determinant 1 |
| 6 | Determinant 6 |
| 7 | Determinant 7 |
| 8 | Determinant 8 |
| 9 | Determinant 9 |
| 10 | Determinant 10 |

Once they completed this initial ranking, another set of instructions was sent to the experts to find a consensus on the most influential determinants. The idea was that they should reflect on the presented averaged rankings based on multidisciplinary knowledge from all experts' responses. Therefore:

1. They had to first answer whether they agreed or not with the variables included in Group A (top-5 averaged determinants averaging the responses from the first phase). We also presented to them a Group B of the least-voted determinants.
2. If they answered positively, they had to rank their personal Group A determinants according to their expert criteria where a determinant placed in the first position meant the best ranking for them and the one in the fifth position was considered the least important.
3. If they had answered in the negative to the previous question, they then had to suggest a new ranking, with the possibility of combining or changing a variable from one group to another (Groups A and B, see Table 3).

**Table 3.** Group A and B creation and associated ranking from the first round of Delphi method.

| Group A | | |
|---|---|---|
| **Averaged order** | **A** | **Description** |
| 1 | A1 | Marketing campaigns generated by automotive brands and dealerships could influence maintenance expenditure decision-making. |
| 2 | A2 | Point-of-sale distance could actively influence drivers' maintenance expenditure decision-making. |
| 3 | A2 | Promotions launched by OEMs and the automotive industry could influence decision-making for vehicle maintenance and expenditures. |
| 4 | A4 | Marketing, communications, and promotions are elements that could actively influence drivers' maintenance expenditure decision-making. |
| 5 | A5 | Onboard technology with warning messages generated by the system could influence car-expense-related decision-making (showing potential failures and maintenance reminders). |
| Group B | | |
| **Averaged order** | **B** | **Description** |
| 6 | B1 | The vehicle's type of use (professional or personal, weekday or weekend) could actively influence drivers' maintenance expenditure decision-making. |
| 7 | B2 | Apps and websites in the field of cars could actively influence drivers' maintenance expenditure decision-making. |
| 8 | B3 | Recommendations from family or close friends could actively influence drivers' maintenance expenditure decision-making. |
| 9 | B4 | Comments and recommendations from friends in the automotive field could actively influence drivers' maintenance expenditure decision-making. |
| 10 | B5 | New transport policies (e.g., 30 km/h zones) and their impact on the environment could actively influence drivers' maintenance expenditure decision-making. |

The information gathered from the experts helped the authors to gain insights regarding which determinant was the most important for experts (ranked in the first position) and which was the least important for most of them; a summary table was created to this extent (see Table 4 in the results section).

**Table 4.** Final Delphi method ranking with the most popular determinants.

| Ranking | Short Description | Determinant | Determinant Full Description |
|---|---|---|---|
| 1 | Driver Artefact System (DAS) | A5 | On-board technology with warning messages generated by the system could influence car expenses decision-making (showing potential failures and maintenance reminders). |
| 2 | Proximity | A2 | Point of sale distance that could actively influence drivers' maintenance expenditure decision-making. |
| 3 | General Marketing | A4 | Marketing, communications, and promotions are elements that could actively influence drivers' maintenance expenditure decision-making. |
| 4 | OEM Marketing | A1 | Marketing campaigns generated by automotive brands and dealerships could influence maintenance expenditure decision-making. |
| 5 | Promotions | A3 | Promotions launched by OEMs and the automotive industry could influence decision-making for vehicle maintenance and expenditures. |

*3.2. Second Phase: Comparative Survey Method*

After using the Delphi methodology, and to complement it, a survey was launched to compare the savings of drivers with connected vehicles with those with non-connected vehicles regarding car mobility scenarios (i.e., the vehicle is in a certain area at a certain time, so it is advisable to suggest a certain action to the driver) and the vehicle itself (e.g., it needs fuel, servicing, or an oil change). This method was used to confirm the findings of the experts in the exploratory phase.

The idea of the survey was to confirm if the determinants driven by the experts are indeed those with the highest impact on real scenarios or not. This survey was part of a global data collection process performed within this research. The whole questionnaire can be consulted in the following link in the author's mother language (https://drive.google.com/file/d/13EIiXGI8Z6F-xexgUc1irUOxOr-9iMZB/view, accessed on 18 January 2022), or in Appendix A.

Hence, a survey was launched in an online format, again using Google Forms, in order to evaluate and compare the spending of drivers of connected and non-connected vehicles. In both cases, the link to the online form was sent by email. Such surveys were sent out to 1500 drivers with connected vehicles, and a snowball sampling technique [35,36] was used to send the same surveys to drivers of non-connected vehicles. As will be shown later, 302 valid responses from users with connected cars and 254 responses from drivers without connected cars were collected. Both sample datasets (for connected and non-connected vehicles) have the same country and the same geographical area; there was no electric vehicle in the sample. This sample is significant since it represents the average age of cars in Spain and includes people with very similar social-demographic profiles in order to avoid any kind of statistical bias. This made it possible to compare the groups and determine the impact of the stimuli received by drivers of connected cars on their expenditures and savings. The authors of this text reached an agreement with the Grupo Next company (www.gruponext.es and www.nextsmartcar.es, accessed on 18 December 2021) to provide them with a sample of 1500 drivers of connected vehicles. This company is one of the European actors in the field of connected cars and offers users different services to make car-related activities less expensive and easier to manage. The completely anonymized sample of drivers making journeys in Spain was extracted from the company's database. In the case of the non-connected cars' drivers, through a snowball technique, we sent emails to different groups of collaborators who subsequently sent the emails to others, and these then sent them to many others

until the number of drivers was like the one corresponding to the connected cars sample. As earlier mentioned, all the vehicles in both samples represent the average car age in Spain (12.7 years) (https://www.lavanguardia.com/motor/actualidad/20210125/6197134/espana-parque-automovilistico-edad-media-coches-viejos-paises-europa.html, accessed on 18 January 2022).

The survey questions were validated through a group of experts before launching the form. The data protection guidelines, the legal General Data Protection Regulation (GDPR), and anonymization policies were first agreed upon by participants before starting the questionnaire. It is important to emphasize that the authors of this paper did not have access to either the private information or the identity of drivers. The respondents were informed about the purpose of the questionnaire and about its scientific use. Finally, users were informed that their participation was voluntary, and some reminders were sent during the survey period to try to ensure an adequate participation rate. Additionally, it should be mentioned that the data were captured and analyzed with the ethical validation of the process defined by the Research Ethics Committee of the University of Deusto (https://www.deusto.es/cs/Satellite/deustoresearch/en/home/research-with-us/research-ethics-comittee/composition, accessed on 18 December 2021).

All the drivers in the connected-vehicle sample had at least 6 months of experience with the ICT-based system and, thus, had experience with a device installed in their vehicle that could keep track of their mobility. This point was very important as the purpose of the study was to measure the impact of the technology on the expenditures and savings of both sets of drivers and their behaviors regarding car maintenance and expense management. Two reminders were sent to those in the selected sample until the collection was considered final.

The survey questions can be consulted after translation in Appendix A. These were prepared based on three criteria: (i) identify the car and its uses, (ii) the expenses around the car and the vehicle mobility, and (iii) the driver perception of vehicle expenses. The rest of the survey questions could be consulted in the aforementioned link.

The first survey section contained four questions to identify the car and its use. The second section contained six questions to understand the expenses around the car and its mobility patterns on a yearly basis. All of them had five response options (a scale of expense ranking and DN/DA "don't know, don't answer". The last section (Section 3) contained two questions related to driver perception of vehicle expenses and had six multiple-choice response options based in a Likert scale: very low, low, regular, high, very high, and DN/DA. The same questionnaire was sent to both samples to avoid any bias. Finally, it is worth mentioning that most of the questions were followed by a confidence criterion. Thus, when we asked drivers about the expenses they incur on their cars on a yearly basis, we afterward asked them to provide a confidence rating from 1 to 10 for their responses, where 1 was poor confidence in their response and 10 was complete confidence in what they responded. We used this confidence criteria to weigh the responses given by drivers when we computed the average expenses by year. Furthermore, it helped us to compare if the drivers with connected cars are more confidence in their responses than those without connected cars. That is, if the technology also plays a role in the acquaintance people have concerning their driver expenses.

## 4. Results

### 4.1. Delphi Results

Following the description and definition of the Delphi method explained in Section 3, the results are presented in what follows.

4.  A total of 24 experts (one of them did not respond at this stage, so we decided to continue with 24 experts instead of 25 in order to not stop the process) responded with their determinants and these were examined, curated, grouped, adjusted, and finally formatted in a table (see Table 1).

5. A total of 22 experts, out of the remaining 24, responded to the questionnaire to reach an initial agreement when it comes to prioritizing which determinants were more or less important. We, therefore, had an attrition rate of three people who did not respond to the questionnaire after several gentle reminders. The results of the first averaged ranking among the 22 experts can be observed in Table 2. Again, the information was input into a personal database of the research team for further analysis.

The determinants from the first consensus round were then clustered into two groups: Group A and Group B. Group A included the five factors that reached more consensus in terms of being the most relevant determinants for car expenditures according to experts' criteria (i.e., Determinants 4, 5, 2, 3, and 1); Group B included the other five determinants, which received fewer votes out of the total 10 determinants (i.e., Determinants 6, 7, 8, 9, and 10). They were presented according to the aforementioned criteria (see Table 3) and presented again to experts for the second consensus stage according to the Delphi method.

All data analysis was conducted on an MS-Excel spreadsheet.

After having presented the list of the determinants grouped in two categories, the group of experts was requested to rank the table again. This process was launched twice to re-confirm the answers (on two different days). First, the group of experts concluded that the five determinants were indeed the most relevant in terms of drivers' level of spending around their car mobility. Second, they ranked those determinants as reflected by Table 4. Hence, Table 3 collects the information from the experts while Table 4 orders and ranks the opinions of the experts. Both tables show the same information, the only difference being that Table 4 is ranked according to experts' consensus decisions.

As shown, the highest ranked (top 1) determinant was "on-board technology offering warning messages generated by the car (IoT device)" regarding driver's expenses according to all experts' opinions, followed by "proximity" and "marketing campaigns".

*4.2. Survey Results*

As explained above, the survey was launched and structured into three sections (See Appendix A): (i) the first section focused on general vehicle information, (ii) the second section focused on consumption and expense information, and (iii) the third section referred to the perception of drivers regarding vehicle costs.

The purpose of the first section was to check whether there were any biases, patterns, or data that might suggest that the sample was not of an adequate size or could contain alterations disrupting the data collection process (e.g., one sample contained vehicles that were much more modern, differences in car age, etc.). As can be seen in Table 5, both samples were similar in terms of types of cars, year of car production, years of car position, and numbers of kilometres per year.

**Table 5.** Comparison of data of connected and non-connected cars (Section 1).

| Question | Connected Car | Non-Connected Car |
|---|---|---|
| Q1-Vehicle Brand | 20 most popular brands in | 20 most popular brands in |
| Q2-Vehicle year (median) | 2014 | 2013 |
| Q3-How long have you had the vehicle? | 5.82 | 5.58 |
| Q4-No. of km/year (mean) | 13,103.82 | 12,786.5 |

The initial step regarding the results of Section 2 was to normalize the data of each sample (connected and non-connected cars) for each question (total of six). To do this, all the values were first translated into an expenditure index. For instance, if we asked about the kilometer per year and the response in the form was "between 10,001 km and 15,000 km", we translated that value to an average scalar (i.e., 12,500 km). These normalisations helped us to calculate average costs per year or the average cost per kilometer. Subsequently, we calculated the descriptive statistics for each of the users' data (Tables 6 and 7). The

confidence level for all dimensions (Q1 to Q6) among both populations is sufficiently high (all dimensions in both sets have a 99% confidence level with a maximum ±15% error, as seen in Tables 6 and 7). Such high-quality datasets enabled us to proceed to get valid and usable conclusions according to the statistical analysis proposed [37].

**Table 6.** Connected car statistical data (Section 2 of the questionnaire).

|  | Q1 (Petrol) | Q2 (Insurance) | Q3 (Mainte- nance) | Q4 (Car Expen- ditures) | Q5 (Recommen- dations) | Q6 (Savings) | ∑ Q |
|---|---|---|---|---|---|---|---|
| Mean | 43.55 | 398.59 | 291.41 | 1516.61 | 32.97 | 55.43 | 1.18 |
| Standard Error | 0.66 | 8.91 | 7.96 | 33.09 | 2.26 | 3.14 | 0.00 |
| Standard Deviation | 11.69 | 157.71 | 140.78 | 585.50 | 39.82 | 55.64 | 0.07 |
| Sample Variance | 136.55 | 24,872.38 | 19,817.89 | 342,816.06 | 1585.62 | 3095.41 | 0.00 |
| Confidence Level (99.0%) | 1.71 | 23.10 | 20.62 | 85.77 | 5.86 | 8.15 | 0.01 |

**Table 7.** Non-connected car statistical data (Section 2 of the questionnaire).

|  | Q1 (Petrol) | Q2 (Insurance) | Q3 (Mainte- nance) | Q4 (Car Expen- ditures) | Q5 (Recommen- dations) | Q6 (Savings) | ∑ Q |
|---|---|---|---|---|---|---|---|
| Mean | 40.18 | 396.63 | 306.06 | 1508.33 | 21.72 | 60.67 | 2.39 |
| Standard Error | 0.97 | 12.85 | 11.56 | 42.96 | 2.32 | 3.83 | 1.19 |
| Standard Deviation | 16.31 | 215.86 | 194.15 | 721.49 | 38.98 | 64.28 | 18.93 |
| Sample Variance | 266.09 | 46,595.37 | 37,693.70 | 520,544.19 | 1519.82 | 4132.46 | 358.46 |
| Confidence Level (99.0%) | 2.52 | 33.34 | 29.98 | 111.42 | 6.02 | 9.93 | 3.10 |

Additionally, the total expenditure level of each of the respondents in each of the samples was compared on a yearly basis. Furthermore, these were calculated according to the kilometers travelled by each user per year (see Table 8). Thus, the differences between the two samples are clearly significant, i.e., a difference of EUR 221.80, calculated by subtracting their corresponding expenditures, EUR 2758 and EUR 2979.80, respectively.

**Table 8.** Total expenses and average expenditures on a year basis presented first without taking into account the confidence level and then by weighting the data according to it.

|  | Connected Cars per Year (*n* = 302) | Non-Connected Cars per Year (*n* =252) |
|---|---|---|
| Total Expenses | EUR 2758.0 | EUR 2979.8 |
| Average expenditure per km travelled | EUR 0.210209 | EUR 0.22752 |
| Average expenditure per km travelled weighted to the confidence level of the given responses | EUR 0.183214 | EUR 0.19555 |

Finally, we averaged the total cost per year with the confidence level they gave to their responses (as can be observed in the questionnaire and as it has been explained above, after each question related to the estimated costs per year, we asked them to rank from 1 to 10 the level of confidence in the answer they provided). The results of the level of expenses associated to each sample are shown in the third line of Table 8. As can be seen, the number of respondents is smaller than the number presented in the introduction. This is because we removed outliers and uncompleted responses in the questionnaire.

To validate that the results from Table 8 are significant from a statistical point of view, we used the t-test with the following results.

There was a significant increase in the overall cost per year and per kilometer for the people with non-connected cars (M = 0.19555, SD = 0.0746) compared to the respondents with an IoT device installed in their cars (M = 0.183214, SD = 0.0676), t(552) = 1.9642, $p < 0.05$. Please note that this parametric test was applied in the responses weighted to the confidence level.

The results of Section 3 were averaged and mapped according to response levels on a Likert scale from 0 to 6, where 1 corresponded to "very low" and 6 to "very high" (0 for "DN/NA"). The results are as shown in Table 9. As a result, people with connected cars seem to have a lower perception of car mobility expenses than those with non-connected cars, whereas this latter result did not offer significant results from a statistical point of view.

**Table 9.** Perception of vehicle expenses (Section 3).

| Perception of Vehicle Expenses | Connected Cars | Non-Connected Cars |
| :---: | :---: | :---: |
| Q1 (% of expenses) | 2.12 | 2.43 |
| Q2 (Promotions) | 2.81 | 2.57 |

## 5. Discussion and Limitations

The experts selected for participation in the Delphi method pointed out in their conclusions that the technology embedded in the vehicle and the warnings and messages it generates for the driver play a decisive role in determining the expenses of the drivers of a vehicle compared to other well-known determinants, such as proximity to the place where the car was bought or marketing campaigns. Based on this conclusion, the survey comparison conducted was suitable for exploring and verifying with real-life testing whether on-board technology could condition the maintenance expenditures around cars or not, and, thus, for determining whether there are positive spill-over effects of having an on-board ICT-based device in the car.

It is interesting to note that the experts were not wrong, since this paper empirically demonstrates that introducing technology into the vehicle reduces the car-related spending. The hypothesis stated at the beginning of the study was therefore confirmed. Drivers who install technology in their vehicles save more (spend less) than those without connected technology in their cars and are more aware about their car expenditures. As a result, it can be concluded that technology plays a key role in saving money and time related to car expenses. The reason drivers of connected cars spend less on vehicle maintenance is because their vehicles' IoT devices will remind them to do maintenance so that the vehicles' engines keep working well and, in consequence, enhance fuel consumption and reduce the possibility of vehicle breakdown. Consequently, drivers will adapt their driving behaviors as induced by the messages issued by the IoT device, reducing car maintenance costs.

A limitation of this study is that the driving styles are not considered. This certainly could influence fuel consumption performance, but not the other factors.

The survey presented and the data gathered showed that drivers of connected cars have a clear tendency to track savings. Users with connected cars either save more or are more aware of savings because of the technology embedded in their cars.

Expenditures on a car are important for people who drive a connected car, but are not so much for people who drive a vehicle without built-in communications technology (because they are simply not aware of what technology can do for them). This leads one to think that the mere fact of having the technology and information makes people with connected vehicles more aware of their vehicle's annual cost.

A major future lever for cars and its associated mobility services will be the adoption of 5G, enabling new services for cars, thereby making mobility-related activities more comfortable and satisfying.

While car mobility has brought enormous value to society over the past 100 years, there is still room to advance, particularly regarding efficiency, mobility cost, and car usage. There is great unexplored potential to study the impact of introducing technology to reduce expenses around cars, such as road infrastructure (free highways or toll highways), driving route information (more or fewer kilometers), and repair shops (with reverse auction). In this sense, technology should be viewed as an essential part of the car, and this technology should be connected to the different product and service providers to matchmake the drivers' real-time needs with the best possible solutions in each case.

It is worth pointing out that connected cars not only help change the user's behavior but also have further advantages since this type of technology is a real-time communication enabler that allows the sender of the message and the receiver to adapt their responses at any given time. In future work, we would like to study user behavior by considering a more sophisticated driver profiling and contextual model, possibly based on semantic ontologies. Determining users' tastes and needs (profiling and classifying them) may help in identifying the products that users need before designing the sale of that product—in other words, by first stimulating demand and then by designing supply. It would, therefore, not be a question of selling products, but of satisfying the needs of individuals.

Before finishing, it is important to indicate the limitations of the research work presented. Age and gender can be two factors that can condition the results and those variables have not been considered; nor has the driving style (the more aggressive it is, the higher the fuel consumption, for example). The socio-economic contexts of the survey participants have not been considered either. The number of people who participated in the survey samples, although significant, could be considered too small to generalize the findings. Furthermore, we assume that the normalization process with the expense data may lead to an average bias, as we translated continuous values into concrete scalars for all the respondents. All these limitations must be considered and included in a subsequent analysis process to delve deeper into the study that is now being presented. Still, we believe that the depth of this study has been sufficient to be able to assert that instrumenting cars with technology can result in reductions of car maintenance and usage expenses, with important side-effects toward reducing accidents or pollution.

## 6. Conclusions

In this paper we have analyzed the role that emerging technologies such as IoT can play in the expenses that drivers usually incur throughout the year, finding that warnings and alerts issued from that technology help to lower the money spent on car maintenance. To obtain this result, a two-stage methodology was used. On the one hand, we ran a Delphi approach with 25 experts in the field of cars to understand the main determinants for car expenses. In this initial stage of the exploratory research, we found that technology has a salient role outperforming, according to experts' criteria, other methods such as proximity to point of sale or marketing campaigns. On the other hand, to confirm that the experts were right, we delivered a survey to two population samples: people with assistive technology in their cars and people with conventional cars without such ICT assets built in, reaching more than 500 people in total. The results from that survey showed that drivers with IoT devices embedded in their cars spend significantly less money on car maintenance (i.e., petrol, insurance, maintenance, car servicing, garages, tolls, penalties, taxes, and repair costs) than drivers without any kind of assistive technology. Moreover, there are signs that emerging technologies in cars can provide more confidence and knowledge to the drivers about the overall car expenses throughout the year. Thus, people are savvier or more aware about the money they spend on their cars on a yearly basis.

The introduction of all these technologies will transform the traffic in our cities, increasing safety, improving smart mobility, and reducing the costs of transportation.

**Author Contributions:** Conceptualization: J.G.-G., D.C.-M., and D.L.-d.-I.; methodology: J.G.-G.; validation and formal analysis: J.G.-G., and D.C.-M.; writing—original draft preparation: J.G.-G., and D.C.-M.; writing—review and editing: D.C.-M., and D.L.-d.-I.; supervision: D.L.-d.-I., and D.C.-M. All authors have read and agreed to the published version of the manuscript.

**Funding:** This paper has been sponsored by grant IT1078-16 associated to DEUSTEK research group associated to the Basque university system.

**Institutional Review Board Statement:** Not applicable.

**Informed Consent Statement:** Not applicable.

**Data Availability Statement:** https://drive.google.com/file/d/13EIiXGI8Z6F-xexgUc1irUOxOr-9iMZB/view (accessed on 18 January 2022).

**Acknowledgments:** We would like to thank all of the expert participants and all those who helped with the survey. We also acknowledge the Ministry of Economy, Industry and Competitiveness of Spain for IoP, under Grant No.: PID2020- 119682RB-I00.

**Conflicts of Interest:** The authors declare no conflict of interest.

## Appendix A

**Table A1.** Driver survey questions.

| SEC 1 | Vehicle Data |
|---|---|
| Q1 | What is the brand of your vehicle? |
| Q2 | What year is your vehicle? |
| Q3 | How long have you had your vehicle? |
| Q4 | How many kilometres do you drive per year in your vehicle? |
| SEC 2 | Mobility and consumption data |
| Q1 | How much money do you spend per month on Petrol |
| Q2 | How much money do you spend per year on insurance for your main vehicle? |
| Q3 | How much money do you spend per year on maintenance to your vehicle (change of oil, filters, tyres and other incidentals)? |
| Q4 | How much money do you think you spend per year on your vehicle (including petrol, insurance, taxes, repairs, maintenance, parking, tolls, fines, etc.)? |
| Q5 | How much money do you think could be saved per year through a recommendation IT embedded system? |
| Q6 | How much money do you think you could save per year on your vehicle expenses by using a recommendation system based on real needs at any given time? |
| SEC 3 | Perception of vehicle expenses |
| Q1 | What percentage of the household expenses is used for vehicle expenses? |
| Q2 | Do you often use promotions or special offers to try to save money on vehicle expenses? |

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
