# Peer review of "The Role of IoT Devices in Sustainable Car Expenses in the Context of the Intelligent Mobility: A Comparative Approach"

_applsci, doi:10.3390/app12031080_

Round 1
Reviewer 1 Report
This paper studies the impact of ICT technologies embedded in vehicles (the so-called Connected Cars) on increasing drivers awareness of car expenses. The paper is interesting and of significant scientific soundness. Still, the paper could be improved in various ways before being accepted:
- Embedded ICT technologies that we commonly found in connected cars are pretty diverse, and use different technologies (V2X communication like DSRC, C-V2X, cellular technologies like 4G, 5G, Android systems, Amazon/Google/Microsoft IoT/IoV solutions). Providing some insights regarding these important aspects can help the readers grasp the significance of this study.
- Excluding the expenses linked to the embedded ICT technologies (purchase + maintenance) when conducting this study creates a bias. While it can be difficult to redo such an important study, authors should at least underline this factor.
- Not all readers are familiar with described methods, namely, a Delphi method. Providing a small description/description could be a real added value for the paper.
- "As shown, the highest-ranked (top 1) determinant was “ on-board technology offering warning messages generated by the car (IoT device)” regarding driver’s expenses according to all experts opinion, followed by proximity and marketing campaigns.": this is confusing and a bit strange to notice such a change between Tables 3 & 4, authors could better explain/justify this.
- The paragraph between lines 419-424 is confusing, and consequently should be revised to be made more clear, easy to read.
- Tables 6 & 7 could be better summarized without providing all these numbers that ultimately confuse readers.
- Tables 8 & 9, as well as Section 5, assess clearly that ICT technologies increase the awareness regarding car expenses, however, this is not trivial in view of the limited differences between the results of each vehicle category. Rephrasing this part, or at least nuancing the findings seems more adequate.
Author Response
Please, see the attachment
Thank you in advance

Reviewer 2 Report
In Methodology:
- A little bit of explanation about using Delphi in this research would be great. Moreover, the method used in the research is a little bit like Fuzzy Delphi (Fig. 2).
- The usage of Delphi during COVID-19 period would have some uncertainty, how did the research design try to cope with?
- Line 277 and 282: Readers need to see the tables in Line 384 and 396. Is that reasonable?
- Line 509-521: if these are research limits, would it be better to put them in Methodology rather than Discussion?
In Results, Discussion and Conclusion:
- Car maintenance includes petrol, insurance, maintenance, car service, garage, tolls, penalties, taxes and repair costs. The range is too wide to indicate the key savings. For example, driving style certainly influences the fuel consumption performance, and it also implies that the drivers have greater opportunities to receive fines due to speeding. Thus, such a result somehow weakens the contribution this research has made.
- Following the last point, the reason drivers spend less in vehicle maintenance is because their vehicles with IoT, but why? Is it because the vehicle will remind them to do maintenance so that vehicles’ engines works well and, in consequence, enhance the fuel consumption so that drivers save money? And reduce the possibilities of vehicle breakdown so that drivers save money? And these behaviours in total save insurance expenditure? Is it because IoT reminds drivers there are speed camera ahead, so drivers slow down and avoid receiving tickets?
- Line 491-498: not sure whether the information is worth mentioned. The focus of this research is the interaction between technology and drivers’ expenditure – saving the planet is another thing.
Author Response

(The authors gave the same response as above.)
